Testosterone improves erectile function through inhibition of reactive oxygen species generation in castrated rats

Li Rui 1 2
Meng Xianghu 1 2 3
Zhang Yan 1 2
Wang Tao 1 2
Yang Jun 1 2
Niu Yonghua 1 2
Cui Kai 1 2
Wang Shaogang 1 2
Liu Jihong 1 2 jhliu@tjh.tjmu.edu.cn
Rao Ke 1 2 raokeke2009@163.com
1 Department of Urology, Tongji Hospital, Tongji Medical College, Huazhong University of Science and Technology , Wuhan, Hubei , China
2 Institute of Urology, Tongji Hospital, Tongji Medical College, Huazhong University of Science and Technology , Wuhan, Hubei , China
3 Current affiliation: Department of Urology, First Affiliated Hospital of Nanjing Medical University , Nanjing, Jiangsu , China
Cui Ranji
Electronic publication date: 2016 May 3
Publication date: 2016
Volume: 4
Electronic Location ID: e2000
Received 2016 Jan 11; Accepted 2016 Apr 12
Copyright: ©2016 Li et al.
Copyright year: 2016
Copyright holder: Li et al.
License: This is an open access article distributed under the terms of the Creative Commons Attribution License, which permits unrestricted use, distribution, reproduction and adaptation in any medium and for any purpose provided that it is properly attributed. For attribution, the original author(s), title, publication source (PeerJ) and either DOI or URL of the article must be cited.
License URL: https://creativecommons.org/licenses/by/4.0/

Keywords: Testosterone, Reactive oxygen species, COX-2, eNOS, Erectile dysfunction

Funding: National Natural Sciences Foundation of China 81200435 This work was supported by grant from National Natural Sciences Foundation of China (No. 81200435). The funders had no role in study design, data collection and analysis, decision to publish, or preparation of the manuscript.

==============================
Testosterone is overwhelmingly important in regulating erectile physiology. However, the associated molecular mechanisms are poorly understood. The purpose of this study was to explore the effects and mechanisms of testosterone in erectile dysfunction (ED) in castrated rats. Forty male Sprague-Dawley rats were randomized to four groups (control, sham-operated, castration and castration-with-testosterone-replacement). Reactive oxygen species (ROS) production was measured by dihydroethidium (DHE) staining. Erectile function was assessed by the recording of intracavernous pressure (ICP) and mean arterial blood pressure (MAP). Protein expression levels were examined by western blotting. We found that castration reduced erectile function and that testosterone restored it. Nitric oxide synthase (NOS) activity was decrease in the castrated rats, and testosterone administration attenuated this decrease (each p < 0.05). The testosterone, dihydrotestosterone, cyclic guanosine monophosphate (cGMP) and cyclic adenosine monophosphate (cAMP) concentrations were lower in the castrated rats, and testosterone restored these levels (each p < 0.05). Furthermore, the cyclooxygenase-2 (COX-2) and prostacyclin synthase (PTGIS) expression levels and phospho-endothelial nitric oxide synthase (p-eNOS, Ser1177)/endothelial nitric oxide synthase (eNOS) ratio were reduced in the castrated rats compared with the controls (each p < 0.05). In addition, the p40phox and p67phox expression levels were increased in the castrated rats, and testosterone reversed these changes (each p < 0.05). Overall, our results demonstrate that testosterone ameliorates ED after castration by reducing ROS production and increasing the activity of the eNOS/cGMP and COX-2/PTGIS/cAMP signaling pathways.

Introduction

Testosterone is overwhelmingly important in regulating erectile physiology through various signaling pathways (Bond, Angeloni & Podlasek, 2010; Chua et al., 2009; Zhang, Melman & Disanto, 2011b). Erectile dysfunction (ED) is a common symptom that can lead to decreased self-confidence, depression and other symptoms that seriously influence the quality of life (Yohannes et al., 2010). A recent study has revealed that testosterone yields many benefits in the treatment of hypogonadism and ED (Yassin et al., 2014). However, the mechanism of how testosterone improves ED is not completely understood.

Endothelial cells produce and release nitric oxide (NO), which induces the activation of soluble guanylyl cyclase and the accumulation of cyclic guanosine monophosphate (cGMP), resulting in the relaxation of smooth muscle and penile erection (Andersson & Wagner, 1995; Burnett & Musicki, 2005; Lue, 2000). Nicotinamide adenine dinucleotide phosphate (NADPH) oxidase, a complex composed of p22phox, p40phox, p47phox, gp91phox, p67phox and a GTPase Rac1 or Rac2, is a crucial enzyme that catalyzes the production of reactive oxygen species (ROS). Recent studies have reported that ROS play a major role in hypercholesterolemia-induced ED and diabetes-related ED pathogenesis (Li et al., 2012; Musicki et al., 2010; Yang et al., 2013). During the process of hypercholesterolemia-induced and diabetes-related ED, increased oxidative stress leads to an imbalance between the limited antioxidant defenses and accumulated ROS, which induces endothelial dysfunction and decreases NO availability. Finally, increased oxidative stress from the NO/cGMP signaling pathway causes pathological ED (Musicki et al., 2010; Yang et al., 2013). Although effects of testosterone on the NO/cGMP signaling pathway have been documented over the last several years, to the best of our knowledge, no comparative studies have been performed on the role of testosterone in ameliorating ROS in castration-induced ED. Hence, we hypothesized that the above-mentioned changes are present in castrated rats and that testosterone improves erectile function by inhibiting ROS production.

Reduction of the cyclic adenosine monophosphate (cAMP) concentration also occurs in ED. Cyclooxygenase (COX) is an important enzyme involved in prostaglandin synthesis; COX-1 and COX-2 are two COX isoforms. COX-1 is constitutively expressed in cells, and COX-2 is expressed under certain anomalous conditions (Wang et al., 2014). Both of these isoforms transform arachidonic acid into prostaglandin H2 (PGH2), which is further converted into prostaglandin I2 by prostacyclin synthase (PTGIS). Then, adenylyl cyclase is sensitized to produce cAMP. This activation causes smooth muscle relaxation and penile erection (Lin et al., 2013). PGH2 can also be converted into other prostaglandins with potent proinflammatory effects. Any factors affecting this pathway and leading to cAMP reduction may cause ED. Perez-Torres et al. (2010) have found that castration influences arachidonic acid metabolism and reduces COX-2 expression in the kidneys of metabolic syndrome rats. Similarly, Lin et al. (2013) have suggested that COX-2-10aa-PGIS gene therapy elevates erectile function following cavernous nerve injury to rats. However, the role of the COX-2/PTGIS/cAMP signaling pathway remains to be elucidated in castrated rats with ED.

The purpose of this study was to determine the effect of testosterone on the erection process in castrated rats. We analyzed the function of testosterone and investigated the molecular mechanisms of castration-induced ED.

Materials and Methods

Castration model and treatment

In the experiment, 40 adult male, 8-week-old Sprague–Dawley rats weighing 200∼250 g were obtained from Tongji Medical College, Huazhong University of Science and Technology. The rats were randomized into the following four groups: control, sham-operated, surgical castration, and castration-with-testosterone-replacement (n = 10 for each group). The castration procedure was as follows. Briefly, the rats were anesthetized with pentobarbital sodium intraperitoneally (40 mg kg−1). A ventral midline incision was created above the scrotum, and the abdominal wall was cut open. The spermatic cord was then separated, and the vas deferens and associated vasculature were identified and separately ligated. Next, the testicles were removed bilaterally. The rats in the testosterone treatment groups received 100 mg kg−1 month−1 testosterone (subcutaneous injection; Zhejiang Xianju Pharmaceutical Co., Ltd., Taizhou, Zhejiang, China) for 1 month immediately after castration (Zhang et al., 2011a). All procedures involving animals were performed in accordance with the guidelines of the Chinese Council on Animal Care and with approval from the Committees on Animal Experiments at Tongji Hospital (Tongji Medical College, Huazhong University of Science and Technology, Wuhan, Hubei, China; ID: TJ-A20131213).

In vivo assessment of erectile function

Erectile function was assessed in all rats after one month of testosterone treatment. The assessments were conducted as previously described (Li et al., 2012). First, the cavernous nerves were exposed and mounted onto stainless steel bipolar wire electrodes, which were connected to an electrical stimulator. The electrical stimulation parameters were as follows: 5 volts at 15 Hz, with a square-wave duration of 1.2 ms for 1 min. Then, a PE-50 cannula (Becton Dickinson & Co., Sparks, MD, USA) was inserted into the left carotid artery to monitor the systemic mean arterial blood pressure (MAP). Finally, a 25-gauge needle was inserted at the crura, connected to PE-50 tubing, and filled with 250 U mL−1 of a heparin solution. Both blood pressure and intracavernous pressure (ICP) were measured continuously using a data acquisition system (AD Instruments Powerlab/4SP, Bella Vista, NSW, Australia). The Max ICP/MAP was recorded for each rat. The animals were sacrificed via injection of 20 mL of air, and the corporeal tissue was immediately collected from each rat. One-third of the sample was fixed in 4% triformol and embedded in paraffin for further use. The remaining tissue was immediately frozen and stored at −80 °C until analysis.

Measurements of plasma testosterone and dihydrotestosterone (DHT)

Immediately after electrostimulation, blood was collected using a PE-50 tube, which was inserted into the left carotid artery, to determine the testosterone and DHT levels. Whole blood was centrifuged at 1,580 g for 15 min at 4 °C. The testosterone level was determined at the clinical laboratory of Tongji Hospital. The DHT level was determined using an ELISA kit (Westang Bio-Tech Co., Ltd., Shanghai, China). The remaining plasma was collected and stored at −80 °C.

SDS-PAGE and immunoblotting

The frozen penile tissues were prepared in ice-cold RIPA buffer containing a protease inhibitor cocktail and sodium fluoride (NaF), followed by centrifugation at 12,000 g for 15 min at 4 °C. Protein concentrations were assayed using a BCA assay kit (Beyotime Institute of Biotechnology, Haimen, Jiangsu, China). In total, 50 µg of protein was loaded onto a 10% sodium dodecyl sulfate-polyacrylamide precast gel and then transferred to a polyvinylidene fluoride membrane. The membranes were blocked for 1 h in a solution of 0.1% Tris-buffered saline and Tween-20 (TBST) with 5% (w/v) bovine serum albumin at room temperature. The membranes were subsequently incubated with antibodies against p40phox(1:500, Bioworld, Nanjing, Jiangsu, China), p67phox(1:1000, Affinity, Zhenjiang, Jiangsu, China), endothelial nitric oxide synthase (eNOS, 1:1000; Abcam, Cambridge, MA, USA), phospho-eNOS at Ser1177 (p-eNOS, 1:1000; Cell Signal Technology, Beverly, MA, USA), COX-2 (1:500; Abcam, Hong Kong, China), PTGIS (1:1000; Abcam, Hong Kong, China) or β-actin (1:500; Multisciences, Hangzhou, Zhejiang, China) overnight at 4 °C. After the membranes were washed three times in TBST for 1 h, they were incubated with a secondary antibody in TBST at room temperature for 1.5 h. Then, they were washed again three times in TBST and analyzed with an enhanced chemiluminescence detection system (Pierce; Thermo Fisher Scientific, Rockford, IL, USA).

Detection of ROS

The rat corpora cavernosa were quickly frozen, cut to a thickness of 8 µm at an optimized cutting temperature, and placed on glass slides. A fresh dihydroethidium (DHE) solution (1 µmol L−1; Beyotime Institute of Biotechnology, Haimen, Jiangsu, China) was topically applied to each tissue slice, and the slices were incubated for 30 min at 37 °C in the dark. Fluorescence images were captured with an Olympus BX51 fluorescence microscope (Olympus Corporation, Tokyo, Japan). Fluorescence intensities were determined using Image-Pro Plus software (Media Cybernetics Inc., Bethesda, MD, USA).

Determination of nitric oxide synthase (NOS) activity

NOS activity in the penile tissues was measured using an ELISA kit (Nanjing Jiancheng Bioengineering Institute, Nanjing, Jiangsu, China) according to the manufacturer’s instructions. The assays were performed in duplicate, and the protein concentrations were detected to normalize the data.

cGMP and cAMP concentrations

The cGMP and cAMP concentrations in the penile tissues were measured using an ELISA kit (Nanjing Jiancheng Bioengineering Institute, Nanjing, Jiangsu, China) according to the manufacturer’s instructions. The assays were performed in duplicate, and the protein concentrations were detected to normalize the data.

Statistical analysis

Parametric data are expressed as the mean ± SD. All statistical analyses were performed with SPSS 15.0 software (SPSS, Inc., Chicago, IL, USA) using one-way ANOVA followed by Bonferroni’s multiple comparison post-test. Intergroup differences were considered significant at a p < 0.05.

Results

Effect of testosterone treatment on plasma testosterone and DHT concentrations

The castrated rats exhibited marked decreases in body weight, plasma testosterone and DHT levels compared with the control rats. Testosterone replacement restored the testosterone and DHT concentrations, but they were still lower than those of the control rats (although this difference was not significant). There were no differences in the plasma testosterone and DHT concentrations between the control and sham-operated rats (Table 1).

Table 1 Body weight and plasma T, DHT levels in the four groups.

Group	Body weight, g	Plasma T (ng/mL)	Plasma DHT (pg/mL)	
	Initial	Final			
Co	224.7 ± 4.7	407 ± 37*#	4.18 ± 0.27*	142.8 ± 15.8*	
So	225.1 ± 2.5	409 ± 26*#	4.06 ± 0.19*	141.0 ± 18.7*	
Ca	224.5 ± 3.4	340 ± 39	0.51 ± 0.09	48.3 ± 6.0	
Ct	225.4 ± 2.0	343 ± 44	3.93 ± 0.12*	136.0 ± 12.9*	
Notes.

* p < 0.05 vs. the castration group.

# p < 0.05 vs. castration-with-testosterone-replacement group.

Data were expressed as the mean ± SD, (n = 6 ∼ 10 rats/group).

Co control

So sham-operated

Ca castration

Ct castration-with-testosterone-replacement

T testosterone

DHT dihydrotestosterone

N number of analyzed samples

Effects of testosterone treatment on erectile function

Figure 1 presents a summary of the Max ICP/MAP ratios for the four groups. The Max ICP/MAP ratio was lower in the castrated group than in the other three groups subjected to 5 V stimulation. Testosterone therapy resulted in a substantial increase in the Max ICP/MAP ratio compared with that of the castration group with electrostimulation (p < 0.05). However, this ratio was still lower than those of the control and sham-operated rats. There was no difference in the MAP among the four groups.

Figure 1 Testosterone treatment increased the Max ICP/MAP during electrical stimulation of the cavernous nerve (5 V, 15 Hz, 1 min).

(A, B) Representative ICP and MAP tracings in the four groups. Bar graph depicting Max ICP/MAP ratio. The data are expressed as the mean ± SD (n = 6 ∼ 8 rats/group). Co, control; So, sham-operated; Ca, castration; Ct, castration-with-testosterone-replacement. ∗p < 0.05 vs. the castration group; #p < 0.05 vs. the castration-with-testosterone-replacement group.

Effects of testosterone treatment on ROS production in penile cavernous tissue

ROS production was detected in the four groups. As shown in Figs. 2A and 2B, castration resulted in a dramatic increase in ROS production (detected by DHE fluorescence), which was attenuated by testosterone. Furthermore, to assess whether castration-induced ROS in the corpus cavernosum is associated with NADPH oxidase, the protein expression levels of the NADPH oxidase subunits were analyzed. Western blot analysis indicated that p40phox and p67phox were greatly increased in the castrated rats compared with the control and sham-operated rats and that they were markedly reduced by testosterone treatment (p < 0.05, Figs. 2C, 2D and 2E).

Figure 2 Testosterone-induced changes in ROS and protein expression.

(A, B) Typical images of DHE in situ staining in corpora cavernosa from rats in the four groups (red fluorescence; scale bars = 100 µm; time of exposure, 600 ms). Red fluorescence intensity was measured using Image-Pro Plus software. (C, D, E) Representative western blot showing p40phox and p67phox expression normalized to β-actin. The data are expressed as the mean ± SD (n = 6 ∼ 8 rats/group). Co, control; So, sham-operated; Ca, castration; Ct, castration-with-testosterone-replacement. ∗p < 0.05 vs. the castration group. ROS, reactive oxygen species; DHE, dihydroethidium.

Effects of testosterone treatment on the NOS/cGMP signaling pathways in penile cavernous tissue

The expression levels of eNOS and p-eNOS (Ser1177) in the corpus cavernosum were measured by western blotting. The p-eNOS (Ser1177)/eNOS ratio was substantially lower in castrated rats than that in the normal control rats. Treatment with testosterone significantly increased the p-eNOS (Ser1177)/eNOS ratio in the castrated rats (p < 0.05, Figs. 3A and 3B). In addition, to confirm the bioavailability of NO, ELISAs were performed to assess the cavernous NOS activity and cGMP concentration. As shown in Figs. 3C and 3D, the cavernous NOS activity and cGMP concentration were markedly lower in the castrated rats compared with the control and sham-operated rats (each p < 0.05), indicating that the cGMP-protein-kinase-G axis mediated this inhibitory effect of NO. Testosterone treatment significantly attenuated the castration-induced reduction in cavernous cGMP and NOS activity (p < 0.05).

Figure 3 Testosterone-induced increase in the NOS/cGMP signaling pathway in penile tissue.

(A, B) Representative western blotting showing p-eNOS (Ser1177) and eNOS expression normalized to β-actin, as well as the p-eNOS/eNOS ratio. (C) The cGMP concentration was detected in penile tissue. (D) NOS activity in the four groups. The data are expressed as the mean ± SD (n = 6 ∼ 8 rats/group). Co, control; So, sham-operated; Ca, castration; Ct, castration-with-testosterone-replacement. ∗p < 0.05 vs. the castration group.

Effects of testosterone treatment on the COX-2/cAMP signaling pathway in penile cavernous tissue

The cavernous COX-2 and PTGIS protein expression levels were determined in the four groups. These levels were significantly lower in the castration group than in the control and sham-operated groups (each p < 0.05), and they were increased after one month of testosterone treatment (each p < 0.05, Figs. 4A, 4C and 4D). Further, the cAMP concentration was significantly lower in the penile tissue of the castrated rats compared with those of the control and sham-operated rats (p < 0.05, Fig. 4B). The testosterone treatment strongly inhibited the castration-induced reduction in cavernous cAMP (p < 0.05, Fig. 4B).

Figure 4 Testosterone-induced increase in the COX-2/cAMP signaling pathway in penile tissue.

(A, C, D) Typical western blot showing COX-2 and PTGIS protein expression normalized to β-actin. (B) The cAMP concentration was detected in the penile tissue. The data are expressed as the mean ± SD (n = 6 ∼ 8 rats/group). Co, control; So, sham-operated; Ca, castration; Ct, castration-with-testosterone-replacement. ∗p < 0.05 vs. the castration group; #p < 0.05 vs. the castration-with-testosterone-replacement group.

Discussion

Testosterone replacement therapy has been widely studied and has been clinically used for treatment of ED. However, the underlying molecular mechanisms of exogenous testosterone administration are not fully understood and are worthy of a detailed study.

ROS play important roles in various diseases, including cancer, obesity, and ED (Fernandez-Sanchez et al., 2011; Raj et al., 2011; Silva et al., 2014), via reactive elements produced by the reduction of O2 with a single electron (superoxide), two electrons (hydrogen peroxide) or three electrons (hydroxyl radical) (Sabharwal & Schumacker, 2014). A recent study has reported that the penile ROS levels are significantly increased and that eNOS/cGMP activities are reduced in diabetes-related ED (Yang et al., 2013). However, no correlative studies have been performed using a castrated rat model. Excessive ROS production or the failure of oxidant cleaning systems can obstruct cellular function through the oxidation of proteins, lipids and DNA (Murphy et al., 2011). In our study, we found that the levels of ROS were obviously increased and that those of the NADPH oxidase subunits p40phox and p67phox were also increased in the castrated rat model. The up-regulation of p40phox and p67phox resulted in increased ROS levels in the corpus cavernosum. Therefore, the increased production of ROS, which are activated by enzymes involved in their shape (especially NADPH oxidase), might be a key mechanism underlying castration-induced ED.

Several studies have revealed that testosterone is crucial for exerting antioxidant effects by decreasing ROS. Hwang et al. (2011) have demonstrated that testosterone supplementation reduces oxidative damage in Leydig cells. However, the effect of testosterone on ROS levels in the corpora cavernosa of castrated rats is still unclear and needs to be clarified. We found that testosterone treatment reduced ROS level and p40phox and p67phox expression and improved erectile function. The decrease in NADPH oxidase led to a reduction in ROS. Thus, preventing the generation of ROS by interfering with the enzymes that produce them, especially NADPH oxidase, may be a more valid measure for combating oxidative stress than eliminating ROS after their formation.

The NOS/cGMP pathway, which is the primary erectile pathway, has been shown to be associated with androgen. A recent study has revealed that low testosterone levels in men are associated with impaired endothelial function and NO bioavailability (Corrigan et al., 2015; Novo et al., 2015). Effects of testosterone on the expression of NOS isoforms have been shown in penile tissue (Lugg, Rajfer & Gonzalez-Cadavid, 1995; Seo, Kim & Paick, 1999; Traish, Goldstein & Kim, 2007). Replacement of 5α-DHT and testosterone has been shown to restore erectile function and NOS expression in the corpus cavernosum of castrated animals (Schirar et al., 1997; Traish, Goldstein & Kim, 2007). However, the manner by which testosterone enhances the activity of the NOS/cGMP pathway is not fully understood. In this study, we discovered that p-eNOS (Ser1177)/eNOS ratio and the testosterone and cGMP concentrations were reduced in the castrated rats and that treatment with testosterone restored these levels. Numerous studies have concluded that increased ROS generation is one of the major causes of decreased NO bioavailability (De Young et al., 2004; Jin et al., 2009). Hence, according to our findings regarding ROS and NADPH oxidase, we believe that treatment with testosterone ameliorates ED by reducing the expression of the NADPH oxidase subunits p40phox and p67phox. These reductions subsequently trigger a decrease in ROS, improvement in endothelial cell function and an increase in NO. Subsequently, these changes lead to an increase in the cGMP concentration and smooth muscle relaxation in the corpus cavernosum.

In males, testosterone is essential for fertility, puberty, sexual motivation, and sexual performance (Heidelbaugh, 2010). Testosterone production is predominantly regulated through the interaction of luteinizing hormone/human chorionic gonadotropin with specific receptors (Catt & Dufau, 1973; ML, 1998), resulting in an increased intracellular cAMP level. Recent studies have indicated that cAMP plays an important role in erectile physiology through the COX-2 pathway (Lin et al., 2013; Moreland et al., 2001). COX-2 and PTGIS, which regulate the production of inflammatory mediators, are key enzymes involved in cAMP activation. Prostaglandin E, the formation of which is catalyzed by COX-2 and PTGIS, binds to pathognostic receptors on smooth muscle and is thought to enable the relaxation of smooth muscle by activating cAMP-dependent pathways. A lack of testosterone decreases the expression of COX-2 and PTGIS, which in turn results in a reduced cAMP level in the corpus cavernosum. Then, the blocking of cAMP-dependent protein kinase (PKA) activation causes dysfunction in the relaxation of smooth muscle and ED. In our study, we found that COX-2 and PTGIS expression levels were reduced in the castrated rats compared with the control rats. Further, the cAMP concentration was lower in the castrated rats than in the age-matched control rats. Treatment with testosterone markedly increased COX-2 and PTGIS expression, as well as cAMP concentration. These results imply that the COX-2/PTGIS/cAMP signaling pathway may participate in another mechanism responsible for castration-induced ED.

Recent clinical trials suggested a significant improvement in sexual function and ED in hypogonadal men with testosterone treatment (Giltay et al., 2010; Hackett et al., 2013; Khera, 2009; Zitzmann et al., 2013); however, the relationship between testosterone and erectile function has not been completely elucidated. In our study, we revealed that testosterone improved erectile function through inhibition of ROS generation in the castrated rats. These findings could initiate a new line of research in penis physiology and may provide a further scientific basis for the use of testosterone in the management of ED in men with testosterone insufficiency. We hope that these results can be utilized to produce novel therapeutic mechanisms for the treatment of hypogonadal ED.

This study has a few limitations. The possible involvement of the COX-2/PTGIS/cAMP signaling pathway in castration-induced ED needs to be further verified. In addition, the effects of testosterone were evaluated over the short-term in our study; and its long-term effects must be assessed in future studies. Finally, the lack of knowledge regarding the long-term effects of testosterone has limited its clinical application.

Conclusions

In conclusion, testosterone reduced ROS production and increased p-eNOS/eNOS ratio in the castrated rats. Further, it activated the COX-2/PTGIS/cAMP signaling pathway and increased cAMP production. In addition, it improved erectile function in the castrated rats under the combined actions of the above-mentioned factors. Therefore, this study presents novel findings that provide insights into the molecular mechanisms of castration-induced ED. Further studies are needed to elucidate the precise mechanisms involved.

Supplemental Information

Data S1 Raw data

Click here for additional data file.

Additional Information and Declarations

Competing Interests

Author Contributions

Animal Ethics

Data Availability

The authors declare there are no competing interests.

Rui Li conceived and designed the experiments, performed the experiments, wrote the paper, prepared figures and/or tables, reviewed drafts of the paper.

Xianghu Meng performed the experiments, prepared figures and/or tables, reviewed drafts of the paper.

Yan Zhang performed the experiments, reviewed drafts of the paper.

Tao Wang, Jun Yang and Shaogang Wang analyzed the data, reviewed drafts of the paper.

Yonghua Niu and Kai Cui performed the experiments, contributed reagents/materials/analysis tools, reviewed drafts of the paper.

Jihong Liu and Ke Rao conceived and designed the experiments, reviewed drafts of the paper.

The following information was supplied relating to ethical approvals (i.e., approving body and any reference numbers):

Tongji Hospital, Tongji Medical College, Huazhong University of Science and Technology Institutional Review Board (IRB ID:TJ-A20131213).

The following information was supplied regarding data availability:

The raw data has been supplied as Data S1.

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
