# Peer review of "Testosterone improves erectile function through inhibition of reactive oxygen species generation in castrated rats"

_PeerJ, doi:10.7717/peerj.2000_

## Round 0.1 · original submission · Major Revisions

Dear Li

Your manuscript was reviewed by three referees carefully. Our decision is "Major Revisions", Please revise the manuscript before re-submission.

Best Regards
Ranji Cui

Reviewer 1 ·

Basic reporting

The manuscript in title of Testosterone improves erectile function through inhibition of reactive oxygen species generation in castrated rats aims to explore the effect and mechanism of testosterone in castrated rats. The authors demonstrated that testosterone could ameliorate erectile dysfunction (ED) after castration by reducing ROS production and increasing activity of the eNOS/cGMP and COX-2/PTGIS/cAMP signaling pathways.

Experimental design

This is a very interesting project and was well conducted.

Validity of the findings

No comments.

Additional comments

The manuscript in title of Testosterone improves erectile function through inhibition of reactive oxygen species generation in castrated rats aims to explore the effect and mechanism of testosterone in castrated rats. The authors demonstrated that testosterone could ameliorate erectile dysfunction (ED) after castration by reducing ROS production and increasing activity of the eNOS/cGMP and COX-2/PTGIS/cAMP signaling pathways. This is a very interesting project and the manuscript was well organized. A revision is needed before it could be accepted for publication.

1. Line 144, please clarify the testosterone concentration is blood testosterone concentration.
2. Fig 1a, please add the x-axis and unit, also the unit for Y-axis for ICP;
3. In figure 2 D, the label for Y-axis is not correct, please update;
4. Fig 3 B, please correct the Y-axis. Also, ratio of p-eNOS/eNOS is enough, no need to show the total eNOS and p-eNOS;
5. Fig 4, please also update the y-axis.
6. The manuscript need an English editing;

Reviewer 2 ·

Basic reporting

Clear, English language used throughout. and Intro & background to show context. Literature well referenced & relevant.

Experimental design

Original primary research within Scope of the journal. Research question well defined, relevant & meaningful.

Validity of the findings

Data is robust, statistically sound, & controlled.

Additional comments

In this manuscript, the author showed that testosterone could improve erectile function via increasing activity of the eNOS/cGMP and COX-2/PTGIS/cAMP signaling in rats animal model.
The PI tried to conclude that replacement of testosterone in castration rat could partially restore the castration-induced erectile function which be of novelty in this field. Still has some question need to be addressed.

(1) It is still not clearly described how and when treat the rat with testosterone after castration? What is the dosage? Did the author monitor the plasma concentration during the testosterone replacement?
(2) What is the concentration of dihydrotestosterone after castration? Since dihydrotestosterone should be the most active form in serum than testosterone.
(3) The raw data of eNOS/cGMP and COX-2/PTGIS/cAMP in all 40 rats should be uploaded as supplementary figures.
(4) The western blot results were not clearly shown. eg. Fig 3A , too many dots, even actin very near the membrane border. Fig2C. Fig 4A
(5) It was better shown the ICP/MAP and eNOS/cGMP and COX-2/PTGIS/cAMP change simultaneously.
(6) Numerous language errors.

Reviewer 3 ·

Basic reporting

No Comments

Experimental design

No Comments

Validity of the findings

No Comments

Additional comments

The manuscript submitted by Li et al describes potentially significant findings related to the effect and mechanism of testosterone in rat castration-induced erectile dysfunction. Testosterone significantly improves the castration-induced erectile dysfunction through an inhibition of ROS generation and an activation of eNOS/cGMP and COX-2/PTGIS/cAMP signaling pathways. These findings may also have an impact on a better understanding of patients with castration-induced erectile dysfunction.

Minor comments:
1. Please discuss the potential clinical significance. What patients should be benefited from treatment with testosterone? For example: hypogonadism, diabetes-related ED, and cancer-related ED, etc. Can we use the treatment with testosterone for ED in patients with prostate cancer after castration therapy?
2. Few recent discoveries and concepts may help in the discussion. For example, following literatures are not cited in this manuscript.
J Sex Med. 2011 Jul;8(7):1865-79.
J Sex Med. 2010 Mar;7(3):1116-25.
Mol Cell Endocrinol. 2009 May 6;303(1-2):67-73.

---

## Round 0.2 · accepted · Accept

I am glad to inform you that your manuscript is accepted.

Reviewer 1 ·

Basic reporting

No more comments.

Experimental design

No more comments.

Validity of the findings

No more comments.

Additional comments

Authors made extensive revision following reviewer’s comment, no more comments.

Reviewer 3 ·

Basic reporting

Authors addressed all of my questions. There are no addition comments.

Experimental design

GOOD

Validity of the findings

GOOD

Additional comments

GOOD